

# Technical note: TRACK*Flow*, a new versatile microscope system for fission track analysis

Gerben Van Ranst[1], Philippe Baert[2], Ana Clara Fernandes[2], Johan De Grave[1]

[1]Department of Geology, Ghent University, Ghent, 9000, Belgium
[2]Nikon Belux, Groot-Bijgaarden, 1702, Belgium

*Correspondence to*: Gerben Van Ranst (gerben.vanranst@ugent.be)

**Abstract.**

We here present TRACK*Flow*, a new system with dedicated modules for the fission track (FT) laboratory. It is based on the motorised Nikon Eclipse Ni-E upright microscope with the Nikon DS-Ri2 full frame camera and is embedded within the Nikon

NIS-Elements Advanced Research software package. TRACK*Flow* decouples image acquisition from analysis to decrease schedule stress of the microscope. The system further has the aim of being versatile, adaptable to multiple preparation protocols and analysis approaches. It is both suited for small-scale laboratories and is also ready for upscaling to high-throughput imaging. The versatility of the system, based on the operators' full access to the NIS-Elements package, exceeds that of other systems for FT and further expands to stepping away from the dedicated FT microscope towards a general microscope for

Earth Sciences, *including* dedicated modules for FT research.

TRACK*Flow* consists of a number of user-friendly protocols which are based on the well plate design that allows sequential scanning of multiple samples without the need of replacing the slide on the stage. All protocols include a sub-protocol to scan a map of the mount for easy navigation through the samples on the stage. Two protocols are designed for the External Detector Method (EDM) and the LA–ICP–MS apatite fission track (LAFT) approach, with tools for repositioning and calibration to the

external detector. Two other tools are designed for large crystals, such as the Durango age standard and U-doped glass external detectors. These protocols generate a regular grid of points and inspect if each point is suitable for analysis. Both protocols also include an option to image each withheld point. One more protocol is included for the measurement of etch pit diameters and one last protocol prepares a list of coordinates for correlative microscopy. In a following phase of development TRACK*Flow* can be expanded towards fully autonomous calibration, grain detection and imaging.

**1 Introduction**

The fission track method, especially applied to the mineral apatite, is one of the most widely used methods in thermochronology and can be used to reconstruct time-temperature (t–T) paths that a body of rock has experienced in the upper crust through time (e.g. Gleadow et al., 1986; Wagner and Van den haute, 1992; Malusá and Fitzgerald, 2019). As such, it has been proven an ideal tool to reveal the thermal histories of rocks and tectonic events for example in numerous studies. The fission track





method is based on the spontaneous fission decay of $^{238}$U (often present as trace element in a wide range of minerals), which leaves a damage trail or fission track into the crystal lattice. Precise assessment of the accumulation of these tracks through time, analogous to "daughter products", such as stable daughter isotopes in most conventional isotopic dating techniques, allows to date when specific minerals passed through the closure temperature (e.g. ~100 °C in the case op apatite; Wagner and

Van den haute, 1992). Together with track length measurements and a kinetic parameter, such as $D_{par}$ or Cl content, track accumulation information can be applied as input for thermal modelling, which allows the quantitative reconstruction and visualisation of the t–T path a sample experienced (Gallagher, 2012; Gleadow et al., 1986; Ketcham et al., 1999, 2007). As fission track analysis today is indeed mostly applied to apatite, we will continue to focus here on the apatite fission track (AFT) method, although in principle, the technical protocols outlined here are applicable to other mineral detectors as well.

The fission track method distinguishes itself from most other methods of geochronology by its main 'sensors': the human eye or image analysis algorithms (e.g. Gleadow et al., 2009; de Siqueira et al., 2014). Being of this nature, the AFT method is known to be tedious and time-consuming as an analyst needs to correctly identify and count thousands of tracks and measure the length and angles of hundreds of tracks at high (500–1000x) magnification (by optical microscope) for each sample. Furthermore, this puts a high schedule stress on the system since every user needs to spend long times at the microscope.

Traditionally, fission track analysis was done by one analyst, observing and counting tracks through the eyepieces of the optical microscope. Length measurements were either performed using a micrometre eyepiece or with a drawing tube and digitization tablet. With the arrival of CCD and CMOS microscope cameras, the eyepieces were gradually replaced by the PC screen. Conventional assessment of the remaining "parent" isotope $^{238}$U occurs via the external detector method (EDM; Hurford and Green, 1982), in which induced tracks are counted in a co-irradiated external detector (ED), often muscovite. Matching of

crystals and their according induced track clouds is done either by repositioning the ED after etching (Jonckheere et al., 2003) or with an automated stage (Dumitru, 1993; Smith and Leigh-Jones, 1985). More recently, direct measurement of the $^{238}$U, rather than the indirect EDM, for fission track analysis can also be done using the LA–ICP–MS approach (LAFT) (e.g. Hasebe et al., 2004; Chew and Spikings, 2015; Vermeesch, 2017). Besides the development of LAFT, new technology also allowed AFT research to progress in terms of automated image acquisition and analysis (e.g. Gleadow et al., 2009, 2019; de Siqueira

et al., 2014).

When regarding the acquisition and analysis process, the fission track method has a high need for very specific protocols and equipment which are not always readily available from microscope producers. It could thus benefit strongly from further implementation of technological improvements and automation of the process. Notably in this regard is the TrackWorks suite from Autoscan Systems, which pioneered in (semi)automated microscopy for fission track research and was first to have

provided a dedicated system commercially (Gleadow et al., 2019). The TrackWorks software package is custom made by Autoscan, with compatibility for Zeiss motorized microscopes. While these dedicated ready-made or custom-made systems thus do exist, it must be noted that today only a limited number of packages are available on the market for the AFT laboratory. This limits laboratories in their choice of microscope equipment and protocols. Furthermore, for such a singular dedicated protocol, the limited number of packages increases the price.



In this paper we introduce TRACK*Flow*, a novel microscope system developed and optimized for the fission track laboratory, based on the Nikon Eclipse Ni-E motorised microscope (Fig. 1) and embedded within Nikon NIS-Elements software. As such, it serves as a Nikon-based alternative for TrackWorks, broadening market diversity. TRACK*Flow* however diverts from simply offering an alternative, as it intends to widen its application field to general microscopy applications in Earth Sciences

and possibly other Material Sciences. In this note we first discuss the rationale behind the microscope system, followed by a description of the system. We then shortly introduce the protocol modules that are contained in TRACK*Flow* at this time and finish with a conclusion and future perspective.

## 2. System philosophy

TRACK*Flow* has the aim of being versatile and to suit the needs of small- to large-scale fission track laboratories. We provide

a number of protocols based on the actual practical steps necessary for AFT analysis, rather than delivering a singular imaging tool with coordinate transformation. Although we provide a standard sample preparation protocol (Fig. 2; Supplementary material), the system aims to be adaptable to other protocols as well, such as for example illustrated in Fig. 3. The TRACK*Flow* system is suitable for the different approaches of AFT analysis, including LAFT and EDM using both mica or DAP (Tsuruta, 2000) as ED (Fig. 3). This also includes different tactics of analysis, such as expert analyst, computer image analysis (e.g.

Autoscan FastTracks, PyTracks (de Siqueira et al., 2018)) or crowd sourcing (e.g. Geochron@home; Vermeesch and He, 2016) (Supplementary material). As such, different preparation approaches can now produce images in the same settings, stored in a single convenient file format (Fig. 4), which can be exported and analysed in different manners. The microscope can furthermore also be equipped for other basic tasks, such as the analysis of thin sections for example. In this regard we step away from the dedicated fission track microscope and aim to move towards a more versatile microscope system which *also*

*includes* dedicated modules for AFT. In this way the high costs of the advanced system can be covered by multiple tasks and different types of users.

The main purpose of TRACK*Flow* is to obtain maximum efficiency from a single microscope system, as such releasing it from schedule pressure, which is often otherwise solved by larger investments into multiple systems. This is mainly done by decoupling image acquisition from the analysis, as one microscope can acquire images that can then be analysed on different

computers. Fission track analyses, which are most time-consuming, can thus be moved to other systems that are not under high schedule stress. Practical for geological samples, it is even more convenient for laboratory-common samples, such as U-doped glasses and age standards. For example, a grid of positions on a Durango crystal can be stored and imaged together with recoordination points (Figs. 5, 8). These are easily recognisable features fixed on the mount, such as a couple of zircons or U-doped glass spherules. These points and their imprints (induced tracks) can be used to transform the coordinates of images to

the stage of another system or an external detector. Sequential irradiations of an age standard then do not require to scan the same positions on the crystal (which are already imaged). Instead the coordinates can be loaded so that only the external detector needs to be imaged. The recoordination points are then used to transform the coordinates so that the locations and

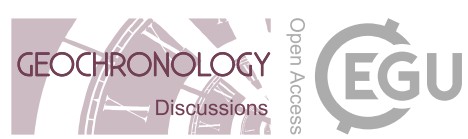

orientation of the images match with the locations on the new ED (Fig. 6). Furthermore, as glasses, age standards and their external detectors are available digitally; there is no reason why a new analyst would require microscope time to obtain a $\zeta$ (Hurford and Green, 1983), $\zeta_0$ (Jonckheere, 2003) or GQR-factor (Jonckheere, 2003).

The focus on image acquisition also comes from the need and nowadays also the possibility to make data available in its most raw form (e.g. FAIR Data Commitment Statement). At the same time, samples are archived for the research database of the institution and can be made available for students. Furthermore, with imaging the tracks are digitally preserved for destructive LAFT analysis.

## 3 Microscope system

The system is based on the Nikon Eclipse Ni-E motorised upright microscope (Fig. 1), with the dedicated modules for AFT research embedded within the Nikon NIS-Elements Advanced Research (AR) software package with JOBS smart imaging design interface. NIS-Elements AR contains, amongst others, well-developed basic microscopy features such as auto-exposure, auto white balance, multiple contrast methods, autofocus and advanced image analysis. Incorporating the TRACK*Flow* dedicated modules within this software package also fits in its philosophy of efficiency and versatility. In this perspective it differs from other packages that only offer a limited number of acquisition, image modification and analysis options, as TRACK*Flow* can access a wide range of advanced microscopy features on top of the ready-made AFT modules. The Nikon Eclipse Ni-E microscope can also be equipped with the Märzhäuser SlideExpress 2 auto sampler, as such transforming it into a system for high throughput acquisition for larger laboratories. With a capacity of 120 76x26 mm or 60 76x52 mm slides this autosampler approach supersedes by far other systems that are based on a special stage.

### 3.1 Motorised components

The Nikon Eclipse Ni-E microscope is developed for its automated capabilities. Motorised components include shutters for both dia- and episcopic illumination, field- and aperture diaphragms, optical zoom, condenser, objective and filter cube turrets. All of these components cannot only be controlled directly by the microscope interface but thus also from the software environment, and as such can be embedded into fully automated protocols.

The motorised nature of the optical path allows for a very reproducible and user friendly way of imaging, working with conditioned optical path settings, named *optical configurations*. For each objective the microscope automatically changes to the optimal field- and aperture diaphragm diameters. Moreover, for every illumination condition required, the software reproduces the same conditions of the optical path and camera settings. This is especially useful when switching between apatite mount and an external detector, which often require different camera settings, or when special techniques, such as circular polarisation (Fig. 7), are used as part of an automated protocol.



Another advantage of software with advanced image acquisition and analysis capabilities having control over motorised components, is that it can make smart decisions and execute actions based on conditions of the image. This means that the software rather than just executing programmed actions, can also take over tasks without intervention of a human operator.

### 3.2 Optical components

The diascopic (transmitted) light path is equipped with the Nikon fly-eye lens, which reduces inhomogeneity of the light source. Remaining inhomogeneity can be corrected for in the software by using shading correction images. Light then passes through the polarizer and the NI-CUD-E motorised universal condenser, which has a numerical aperture of 0.88 and houses several positions for additional optical components. In the TRACK*Flow* system, these include a 2–4x auxiliary lens and a custom quarter-lambda plate.

The objective turret houses six positions for Nikon CFI60 infinity optics, containing 2.5x CFI Plan EPI (NA 0.075), 5x (NA 0.15), 10x (NA 0.30), 20x (NA 0.45) and 50x (NA 0.80) CFI TU Plan Fluor EPI and 100x (NA 0.90) CFI TU Plan Apo EPI objectives. All these objectives are of the highest optical standards and are thus ideal for AFT analysis. Especially the 100x Plan Apo objective, which has high chromatic and spheric correction, is necessary for accurate length measurements and to be able to distinguish even the smallest tracks. Naturally, objectives can be changed or added according to the laboratory

preference.

The motorised cube turret contains an EPI-Brightfield, an EPI-DIC cube and a custom cube containing a quarter-lambda plate and an analyser.

The Ni-E microscope is further equipped with the NI-RPZ-E Motorised DSC zooming camera port, which performs a continuous optical secondary magnification of 0.6 to 2.0 times (Fig. 11).

As light source the choice can be made between LED or conventional halogen bulbs if this is required for other applications of the microscope. LED has several advantages over halogen, including a long lifetime, constant light colour for different voltage and through time and a negligible heat production. On the other hand, due to its narrow spectral peak, white light LEDs cannot be used for fluorescence microscopy, which generally requires a different wavelength.

### 3.3 Camera

As the camera has largely replaced the function of the eyepieces, it carries high importance in terms of image quality, contrast and speed. Image acquisition is performed by the mounted Nikon DS-Ri2 colour camera. The 36.0 x 23.9 mm CMOS full frame sensor contains 16.25 (4908 x 3264) megapixels of 7.3 µm pixel size.

Besides of maintaining a high resolution over all magnifications (2.5–100x objectives), this camera produces high signal-to-noise images. This allows to increase analog gain without introducing much noise rather than slowing down image refresh rate

with longer exposure times (Fig. 7).



### 3.4 Stage and focal mechanism

Especially for AFT, high precision and reproducibility of the XY stage and focal mechanism are required.

The standard configuration comes with the Nikon NI-S-E motorised XY stage, which has a resolution of 0.1 µm.

The focal mechanism is of the focusing stage type (opposed to focusing nosepiece type) and has a resolution of 0.025 µm. It

has a built-in linear encoder to guarantee correct readout of the z-axis. The z-drive is thus well-suited for non-horizontal confined track measurements. Together with the camera and software, the focal mechanism can perform autofocus at different programmable settings. It can occur that a small deviation exists between objectives, which is compensated for in the software. Thermal expansion and consequential focal drift are sometimes quoted as a potential weakness of modern microscopes (Gleadow et al., 2019). The Nikon Eclipe Ni-E fitted with two 100 W halogen bulbs has a shift of the focal plain of ~15 µm

during six hours, of which the highest drift rate during the first hour (0.122 µm min$^{-1}$ or 2.0 nm s$^{-1}$). During the following hour the focal drift decreases towards 0.14 nm s$^{-1}$. Given that a deep z-stack of 8 µm depth with 200 nm steps takes approximately five seconds, the uncertainty on the z-axis during the first hour would amount to 0.01 µm. When plugging this value into a track length dip correction, the earliest non-zero second decimal propagates from 11 ° dip (uncorrected for refraction). For z-depths of 5 µm, the error never becomes larger than 0.01 µm. In a track length population suitable for inversion modelling,

this error becomes negligible. Considering image acquisition for track counting, the most important aspect is to capture imagery from around the optimal focal plain of the grains (and ED). As an autofocus is performed for each grain, the effects of focal drift are also well handled by the system. As the here reported focal drift is measured with two halogen bulbs at full intensity, it is reasonable to assume that focal drift will be far less to negligible if these bulbs are replaced by LED.

### 4 Protocols

The Nikon NIS-Elements AR software package is already equipped with advanced features for general microscopy. Nonetheless, the AFT method is very specific and thus requires specialised protocols. All protocols of TRACK*Flow* are created using the JOBS module of the NIS-Elements package. JOBS offers a user-friendly environment in which the operator can adapt protocols or protocol settings according to preference. The protocols are all based on the practical workflow for AFT and include a large degree of adaptability.

Each of the main protocols are a group of similar sub-protocols. Before each operation the operator can choose which *flow* is required. It is possible to let the system send an e-mail when a protocol is finished or when an action is required. An overview flowchart of all imaging protocols can be found in the TRACK*Flow*Chart in Supplement. Figure 9 displays a condensed version of the flowchart.

### 4.1 Well plate approach

Automation is most efficient when part of the protocol can be standardised, such as the locations of interest on a microscope slide. As such the automated microscope has a primary indication and knows where it needs to 'look' for apatite grains or



mount-ED homologous points. Microscope slides with such standardised positions or wells are frequently used in biosciences, where they are known as well plates. In this view, each apatite mount or external detector occupies the place of a well, the microscope slide on which they are attached serving as a well plate.

Although wells are often physical pits, the well positions here are simple markings on a glass slide, which can easily be made

with a stencil, or obtained from the fission track laboratory of Ghent University (Fig. 2). TRACK*Flow* comes with a standard well plate for four EDM mounts and four EDs and a well plate of either two or three wells for LAFT mounts. It is however still possible to generate new well plates, fitting the standard protocol of different laboratories. Geometrical limitations of the wells are set by the XY-limits of the stage and the diameter of the high-magnification objective lenses in case a thin ED is placed on the stage at the same time as the 1 mm thick mount. The problem of difference of thickness can be solved by fixing

small 1 mm thick glass slides on the main glass slide at the position of the EDs.

### 4.2 List operator

The list operator is a small tool to convert between operator preference styles of coordinate storage. The function merges or splits lists of points stored in xml or csv format. Mainly it involves recoordination points and the target grains or positions. The tool also provides the means for coordinate flipping as preparation for correlative microscopy for when the other stage

has different X and/or Y directions.

### 4.3 Transformation engine

One of the main features in TRACK*Flow* is coordinate transformation or micro-georeferencing. Coordinate transformation is most of all necessary to accurately link the positions of apatite grains with their according induced track clouds but is also essential for correlative microscopy, i.e. when the sample is transferred to a different machine with a different coordinate

system, such as a laser ablation cell or SEM. It is also convenient to accurately retrieve points after removing the slide of the stage, e.g. for step etching or other experiments with repetitive imaging.

TRACK*Flow* makes use of a basic 2D Helmert transformation with a least-squares calculation. The Helmert transformation includes a rotation, a translation and an isotrope scaling factor. As such the Helmert transformation can cover most coordinate differences that are met in the application. Shearing of samples is not expected. A minor X- and/or Y-scaling may be however

necessary if the surface of a mount or mica is tilted. In our experience, if care is given to the preparation of plan-parallel mounts, scaling due to tilting is mostly negligible, avoiding the need for more complex transformations.

From classical georeferencing we learn that more points often lead to better transformations. For this reason TRACK*Flow* gives the operator the option to select up to ten homologous point couples, which can be stored for later use. As it is more ambiguous to find homologous points between a mount and an ED, the system gives the average X- and Y-residuals as a metric

for the transformation parameter quality. The system offers the option to perform a secondary selection of homologous points, which is often used when working with EDM and pinprick points (Fig. 5). The primary calibration, based on the coarse pin punctures is then used as a guide to find homologous points of manually pre-selected or randomly selected finer calibration



points, such as zircons or apatite grains. In this case images are acquired from the fine points on the mount, which allow the operator to accurately select the homologous position inside the induced track cloud. We state however that grains with U-zonation or low-U content can still be problematic for this secondary calibration approach.

The order in which two homologous sets of calibration points are selected can be arbitrary, so the operator is not required to
retain a same order. This is due to the sorting algorithm in TRACK*Flow* that is executed before the calculation of transformation parameters. All points are first sorted according to a scanning line that crosscuts the barycentre of the calibration points and rotates in counter clockwise direction (Fig. 5). As this line segment starts horizontally in the direction of the x-axis, calibration points should not be chosen in the middle left section of the mount. On the ED the scanning sweep is reflected horizontally. Allowing an arbitrary order of selecting points is also convenient for when these points are retrieved through
image analysis.

### 4.4 Scout pre-protocol

Sometimes it is convenient to inspect the mount(s) before scanning the grains. The scout pre-protocol can be enabled at the start of each scanning protocol to explore the sample area. The tool herewith provides a large image zoomable 'map' of each mount of the slide or well plate, which can be used to navigate each sample (Fig. 8). A detailed section at the current optical
configuration can be added to this map at any moment. Double-clicking on the XYZ Overview moves the stage to the designated position, such as a certain grain or calibration point. Positions that are marked as targets during a protocol are also indicated on the XYZ Overview map (Fig. 8).

### 4.5 Imaging protocols

### 4.5.1 Durango tool

The main function of the Durango tool is to create a regular grid of points inside large apatite crystals, such as the Durango age standard (Fig. 8). These points represent digital, non-overlapping fragments of the grain, i.e. samples from the entire extent of the grain. The Durango tool first scans a large image of the mount and calculates the extent of the bounding box around the crystal. Optionally this bounding box can be set manually. The system then creates a regular grid inside the bounding box (Fig. 8). The density of the grid can be set manually, based on the number of generated points. If the set spacing between points is
too low, which would cause overlap of images to occur, the system resets the spacing to prevent overlap and prompts a warning. At this point the operator can choose to manually inspect whether the generated points are in epoxy or in the crystal, or start the automatic inspection. The automatic inspection grabs an image in circularly polarized light (Fig. 7) of each point and through image intensity analysis decides whether it is epoxy (dark) or apatite (light). Only points in apatite are retained and exported as a list. This makes that zones crosscut by a large fracture and which are unsuitable for analysis are also not retained
(Fig. 8). Optionally the operator can enable imaging, so that a z-stack image is acquired of each apatite point immediately after inspection. The Durango tool also lets the operator select a number of recoordination points, so that the same grid can be used



for a second scan or for transformation to the ED. Imaging of the ED can then be done using the EDM_s tool (Sect. 4.4.3) with apatite imaging disabled. The exported list of calibration points and generated target points can be imported into the EDM_s tool.

### 4.5.2 Glass tool

The glass tool is similar to the Durango tool, though optimized for the EDs of dosimeter glasses used in the EDM approach. A main difference is that the bounding box around the induced tracks cannot be determined automatically and must be set manually. The glass tool then also creates a regular grid with density of choice (within the limits allowed by overlap protection) and inspects whether there are induced tracks present at the location of each generated point. Only points with induced tracks are retained for export. With imaging enabled, the microscope captures a thin z-stack image of each retained point. It is also

possible to include recoordination points for each glass ED. As such, each generated grid of points inside the glass region can be reloaded for the scanning of the same glass EDs in other irradiation packages. Coordinate transformation is for this purpose included in the glass tool.

### 4.5.3 Mount-ED (EDM) tool

The mount-ED or EDM tool comes in two versions: single mount (EDM_s) or multi-mount (EDM_m). The only difference

between the two is that the single mount version is designed for the scanning of one mount and ED (mounted on the same or different glass slides) at arbitrary locations, and the multi-mount version is intended for mounts and their EDs mounted on a well plate. The latter can thus scan up to four samples in a single run.

The EDM tool first guides the operator through grain selection and the selection of coarse calibration points (usually pin punctures through the ED into the apatite mount). However, the selection can be skipped and existing lists of target points and

calibration points can be imported as well. If secondary calibration is enabled, the operator can then manually choose a number of grains for fine calibration or enable automatic random selection of a chosen number of grains from the target list. The system will then make an image of these fine calibration points at 200x magnification and bring the microscope stage to the according positions on the ED, based on the transformation parameters obtained from the coarse calibration points. The operator is then asked to correct the stage position if necessary. The images of the fine calibration points can be used as a guide for this purpose.

Based on the fine calibration the microscope will obtain z-stack images from all target apatite grains, followed by their according positions on the ED. The images are then available in two separate nd2 files (Fig. 10). It is also possible to disable imaging and to only export the grain and/or transformed ED coordinates.

### 4.5.4 LAFT tool

The LAFT tool is created for the imaging of apatite (or zircon) grains before and after LA–ICP–MS analysis of the mount. For

new samples it first guides the user through the selection of recoordination points and target grains. For existing lists the protocol first asks to import the data, either as separate files for recoordination mark locations and targets or as a single file in



which recoordination marks are stored as the first set of points. This tool is not limited to mounts for LAFT and can also be used to compensate for small differences between slide position when reloading a slide on the stage, such as for EDM. The system then acquires z-stack images of the selected or loaded targets.

*Remark on thick mounts:* It is often to the preference of laboratories to keep mounts at a higher thickness (5 mm). Although
the option to analyse these thick mounts is embedded in TRACK*Flow*, we do not encourage their use due to a number of disadvantages. The first main disadvantage is found in the larger light absorption of the epoxy, leading to either a longer exposure time or higher analog gain. While longer exposure times increase scanning duration, higher gain introduces more noise. A second disadvantage is found in the too distant position of the object relative to the condenser, as it starts to deviate from the 2.50 mm object recommended distance towards the condenser lens surface for proper Köhler illumination. As a result
the best image quality can no longer be obtained as the field diaphragm needs excess opening to prevent a spherical intensity gradient.

### 4.5.5 $D_{par}$ measurement tool.

For thermochronology modelling a kinetic parameter is required to account for the effect of mineral chemistry on track annealing (Carlson et al., 1999; Ketcham et al., 1999, 2007). One approach is to use the etch pit diameter parallel to the c-axis,
termed $D_{par}$. As these features typically are small and in the order of 1 to 3 µm length (5.5 mol L$^{-1}$ HNO$_3$, 21 °C, 20 s etch), measurement accuracy and reproducibility are a challenge. The $D_{par}$ tool provides a means to measure large numbers of $D_{par}$s (dependent on track density) at reproducible imaging and contrast settings. The $D_{par}$ measurement tool takes images of predefined positions in episcopic light at 2000x magnification. It then uses image analysis to separate and measure etch pit diameters, both along ($D_{par}$; maximum Feret diameter) and perpendicular to ($D_{per}$; minimum Feret diameter) the c-axis (Fig.
11). As only intensity and shape parameters are used, the tool however also measures non-FT features, introducing some degree of error. Practice and comparison to non-automatic image analysis however point out that the error due to false features becomes negligible due to the measurement of large numbers. The tool is thus most successful in moderate to high track density samples. For samples with lower track densities or large numbers of defects and/or artefacts we recommend to use the predefined $D_{par}$ image analysis recipe under operator control. This approach also presents the operator with an image overview
of all measured features, which can be sorted. Non-FT features can as such be easily removed from the list of measurements. In other cases some etch pits are missed (Fig. 11). In most occasions this is because these features are filtered out because of their proximity to other tracks or other dark regions. As this filtering is thus random and not systematic and since the goal is to sample a representative $D_{par}$ rather than a quantitative analysis, neglecting of small numbers of tracks does not pose a problem. As an example, for an annealed Durango with induced tracks we obtained a median $D_{par}$ of (1.45 ± 0.02) µm from
103 automated measurements in three spots (5.5 mol/L, 20 °C, 20 s etch). Fifteen manual measurements on the same locations resulted in a median $D_{par}$ of (1.44 ± 0.03) µm.

Although this tool also automatically counts the number of withheld $D_{par}$s measured, it cannot be used for automated fission track counting in its current state. This is mainly because the tool only uses episcopic light, as such missing important





information contained in the diascopic light. Furthermore, automated counting requires a more sophisticated approach (Enkelmann et al., 2012; Gleadow et al., 2009), which is currently not included in TRACK*Flow*. It must also be noted that the $D_{par}$ tool uses images at 2000x magnification to optimise length measurements, as such decreasing the number of counted tracks, which enlarges the uncertainty for counting statistics.

## 4.6 Analysis

When imaging apatite grains, a thick (5 µm) z-stack is acquired in diascopic light and a thin (2 µm) z-stack is captured in episcopic light. For the mica only a thin z-stack is captured in both episcopic and diascopic light. All images of a sample are stored in a multi-dimensional Nikon nd2 file (Fig. 4). This practical file format allows to browse through grains in both illumination types and to scroll through the captured focal levels in a single window. The ED images are stored in a second nd2 file, which can be opened in parallel and can be linked to the window containing the apatite grains (Fig. 10). As such, moving to a next grain image will also move the ED window to the according image. Although this can be conveniently used for alternating (mount–ED) track counting, we recommend to only use the parallel windows to check the state of the images (strong internal reflections in uncoated samples, autofocus difficulties, ...) and to count serially to prevent counting bias (O'Sullivan, 2018). The nd2 files also contain metadata, including the stage location and microscope and camera settings, allowing to revisit the grain and reproduce the same image settings. The image files can be opened in all Nikon NIS-Elements packages (including the free NIS-Elements Viewer) and FIJI (Schindelin et al., 2012). However, most convenient tools for analysis are available in Nikon NIS-Elements AR and Basic Research (BR). Here it is possible to draw one or more regions of interest (ROI) in which the counting will be performed (Figs. 4, 10). The area of the ROI can be measured and it can also be copied to the ED. Both track counting and measuring can be done manually, after which the results can be exported to Microsoft Excel. The operator can also create her/his own recipe for automated image analysis. The resulting separated images of each potential track can then be verified in a list.

The nd2 files are also easily converted to tiff (for external image analysis) or jpeg (for crowd sourcing) as such broadening the possibilities of the analyst.

## 5 Conclusive remarks and further developments

The TRACK*Flow* system has several advantages over other systems that are currently available. First of all it comes with multiple user-friendly modules for different sample types (EDM, LAFT, Durango, glass EDs) and different degrees of automation based on the practical steps for AFT research. Due to the user-friendly design that does not require expert knowledge of AFT, the system can be operated by supporting technicians. The protocols are also easily adapted to the specific needs of the laboratory, giving it a high degree in versatility. In this regard the image material resulting from different preparation protocols can be standardised to the same imaging conditions and stored as one convenient, multi-dimensional format including imaging metadata.



Secondly, the dedicated modules for FT research are embedded in a single software package, the Nikon NIS-Elements package. This increases versatility and adaptability as the software contains a large range of functions that are not available in other packages, broadening the options of the operator. For example, the image analysis function, which contains a wider set of pre-processing and post-separation filter parameters, can be used to create custom laboratory image analysis recipes. Furthermore,

with the JOBS smart imaging design interface it is possible to further expand the existing protocols to the analyst's preference and specific laboratory settings and protocols. The work lay-out is also fully adaptable to the taste of the laboratory or each user. All imagery can be stored in the Nikon Image Database or High Content Database. As microscope control is native and all basic functions such as autofocus are part of the NIS-Elements package, these subjects are covered by the support of Nikon. The system is also the first for FT research to be ready for upscaling to a high-throughput scanner. Due to the modular setting,

upgrading the system is most convenient and can be done rather than purchasing a different microscope. This all makes TRACK*Flow* the first system with dedicated modules for AFT research with such a high degree of versatility and adaptability. Furthermore, rather than being a 'fission track microscope', the system can also be used for other subjects in Earth Sciences, such as the scanning and archiving of thin sections and pre-selections for correlative microscopy.

The system presented here is the current working version of TRACK*Flow*. Continuing developments are however still to be made for a next version of the system. These mainly focus on a more extensive automation, further upscaling and flexibility. Further automation is achieved by training image analysis recipes to be applicable to different samples. These recipes can then be embedded into the current protocols and enable the system to make decisions without intervention of the operator. The main features are the automatic recognition and registration of copper TEM grid central marks (Fig. 5), other recoordination marks

and the recognition of apatite grains. In case there is a large fracture, the system should be able to select the largest area of the grain. For EDM the system can also be trained to detect coarse calibration points in the apatite mount and the ED, such as needle puncture points. If these coarse marks can be provided in a more regular shape (such as U-doped glass spherules), it could also be used as primary fine calibration, making secondary calibration on apatite grains obsolete. For in case a secondary calibration is still used, the system should be able to link the barycentre of grains and their induced track clouds, if the induced

track density is high enough. Once image analysis is embedded in the protocols, these protocols can be merged. In this case marks or a barcode can be provided so that the system can decide without user intervention which protocol it should run. The Nikon Eclipse Ni-E can also be upgraded to a high-throughput system by combining it with the Märzhauser SlideExpress 2 system. The SlideExpress is capable of fitting 120 standard 76x26 mm glass slides or 60 large 76x52 mm slides. In the TRACK*Flow* standard configuration four EDM mounts and four external detectors can be attached to one glass slide. This

allows a sequential digitization of 240 mount-ED couples. For LAFT, mounts of maximum 2 mm thickness can be fit into adapter plates, allowing sequential digitization of 240 samples. This system can thus strongly increase laboratory throughput and can lead to the set-up of imaging centres for example for petrography.
TRACK*Flow* also sets a step away from the 'dedicated fission track microscope' as the combination of the versatility and efficiency of the Nikon Eclipse Ni-E allow to use the system as a general optical imaging system for Earth Sciences. More



precisely, the microscope can be upgraded to a petrographic scanning microscope with rotatable stage simulation. Furthermore, since coordinate transformation is a key tool of TRACK*Flow*, the system can be used for correlative microscopy for e.g. SEM, FIB-SEM or laser ablation (mapping).

## Author contribution

Philippe Baert: Technical support, Software support, writing – reviewing & editing.

Ana Clara Fernandes: Technical support.

Johan De Grave: Supervision, funding acquisition, resources, writing – reviewing & editing.

## Acknowledgements

This work was supported by the Special Research Fund of Ghent University (BOF 01N03915). Our gratitude goes out to Simon
Nachtergaele, who performed the first tests of the protocols both during the development and in their final version. His contributions have helped us to improve the intuitive and user-friendly nature of the workflow.

## Availability

TRACK*Flow* is licensed from Ghent University to Nikon. The full software package for TRACK*Flow* is based on the NIS-Elements AR software suite (Nikon) and can be purchased through Nikon.

## Conflict of interest

The authors declare that they have no conflict of interest.

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

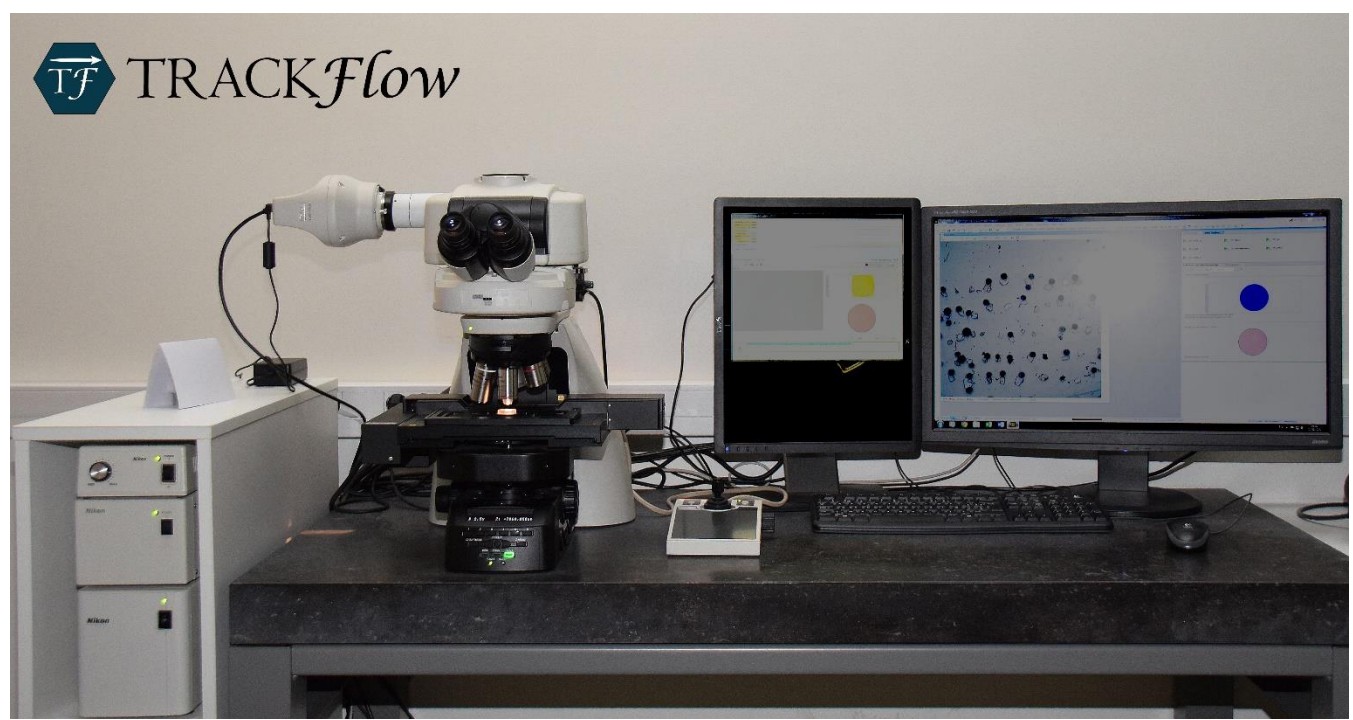

15   **Figure 1: Nikon Eclipse Ni-E microscope with DSC zoom body and Nikon DS-Ri2 camera, configured for TRACK*Flow*.**





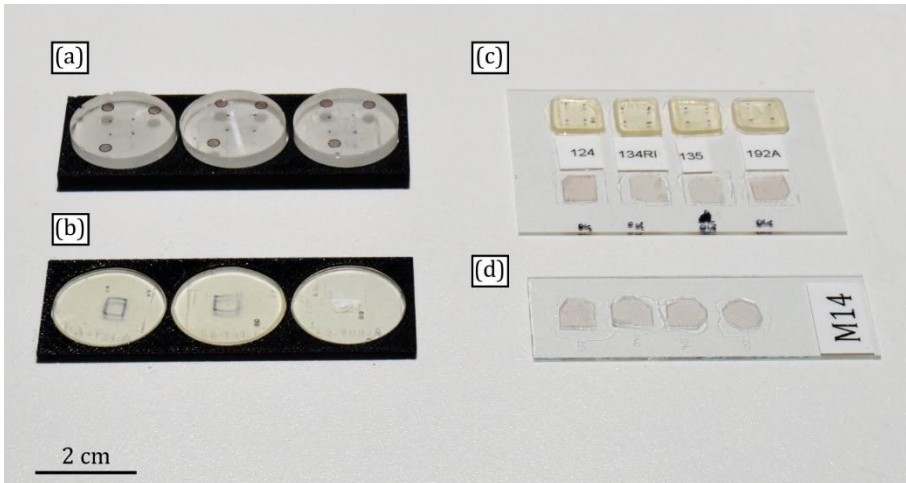

**Figure 2: Standard TRACK*Flow* microscopy slides. (a) Thick 3-mount adaptor slide for thick (~0.5 mm) 1" mounts, usually for LAFT. (b) Thin 3-mount adaptor slide for thin (1 mm) 1" mounts. One-inch (or 25 mm) mounts can be placed in the adaptor slides for imaging and are removed after. (c) Standard EDM TRACK*Flow* (52x76 mm) glass slide. It is possible to raise thin micas to the**
5 **level of the apatite mounts by attaching a standard microscopy glass slide (26x76 mm) to the bottom of the main slide using nail polish. (d) Standard EDM TRACK*Flow* (26x76 mm) glass slide. Used for U-doped glass EDs or separate mounting of apatite mounts and EDs.**

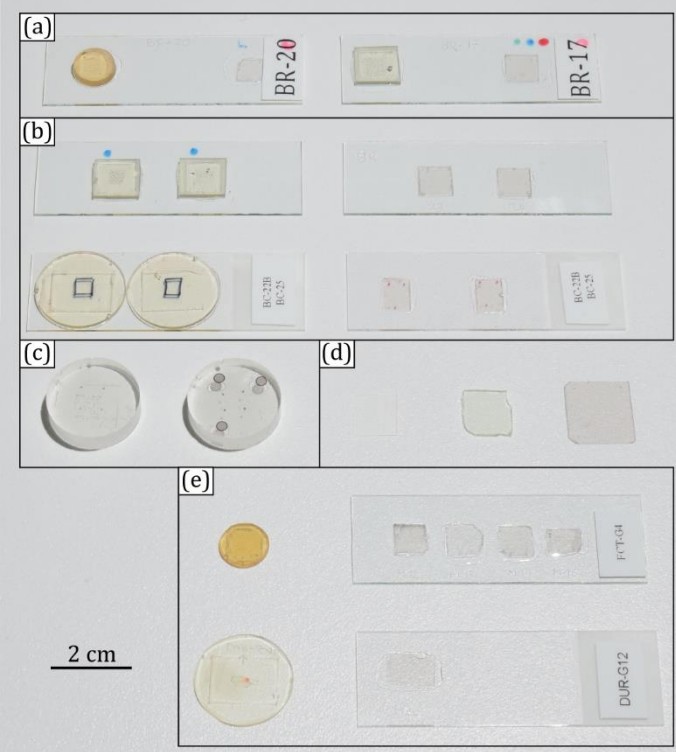

**Figure 3: Overview of different preparation protocols. (a) Mount and mica ED fixed on a same (76x26 mm) glass slide. (b) Multiple**
10 **mounts on a single (76x26 mm) glass slide and EDs on a different glass slide. (c) Thick mounts, usually for LAFT. (d) Three different**

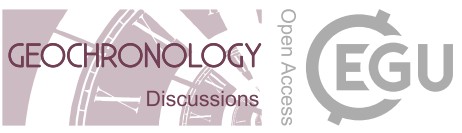

kinds of ED: (from left to right) Bayer Makrofol, DAP (Tsuruta, 2000), Goodfellow natural muscovite mica. (e) Separate mounts and attached EDs. Multiple EDs of a single standard after multiple irradiations.







**Figure 4: Print screen of a Nikon multi-dimensional nd2 file opened in Nikon NIS-Elements BR. a) Look-Up Table (LUT) non-destructive image enhancement settings. b) Synchronisation settings to link different files, such as mount–ED, pre-laser–post-laser**



**or step etching, c) general options, including graticule settings, scale bar and Region Of Interest (ROI) settings, d) navigation through z-stack, e) navigation through XY-positions (grains), f) change between illumination types.**

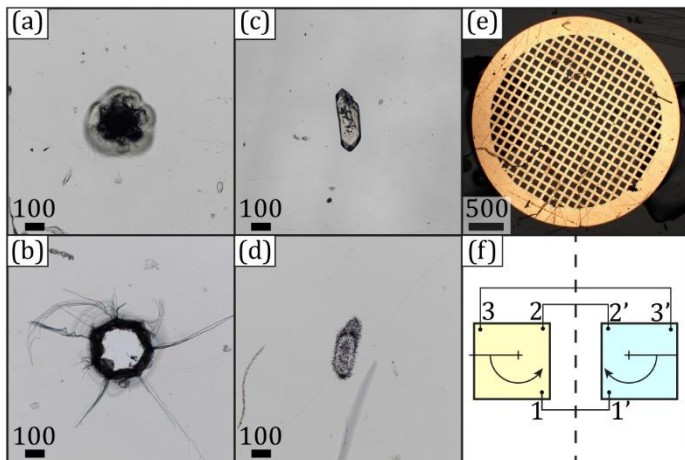

5    **Figure 5: Example of calibration / recoordination marks. a) Pinprick in epoxy mount, b) pinprick in mica ED, c) zircon embedded at a corner of the region of interest in an epoxy mount, d) dense induced track cloud of zircon d in mica ED, e) copper grid on an epoxy mount for correlative microscopy. f) Illustration of microgeoreferencing with automatic sorting. Homologous points are marked in arbitrary order on a mount (left) and ED (right). The points are sorted following the rotating scan line. Mount and ED do not need to be on a single glass slide.**



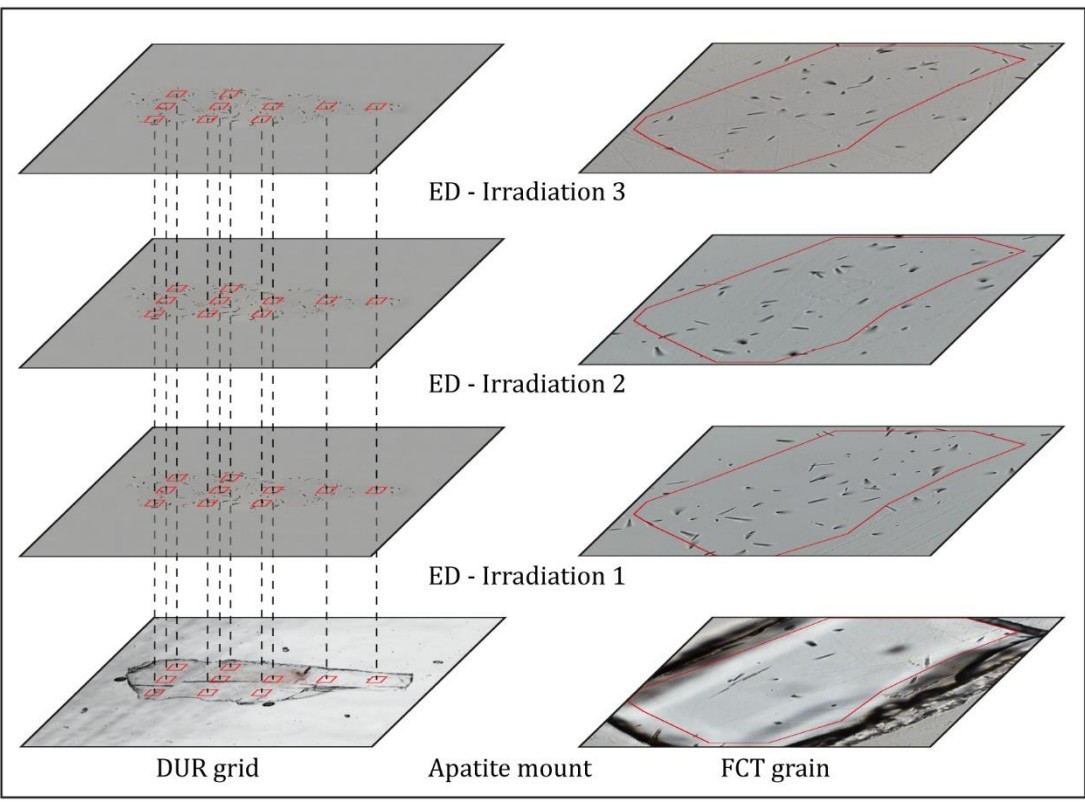

**Figure 6: Standards after different irradiations. Each ED layer corresponds to the same location on the apatite mount layer.**



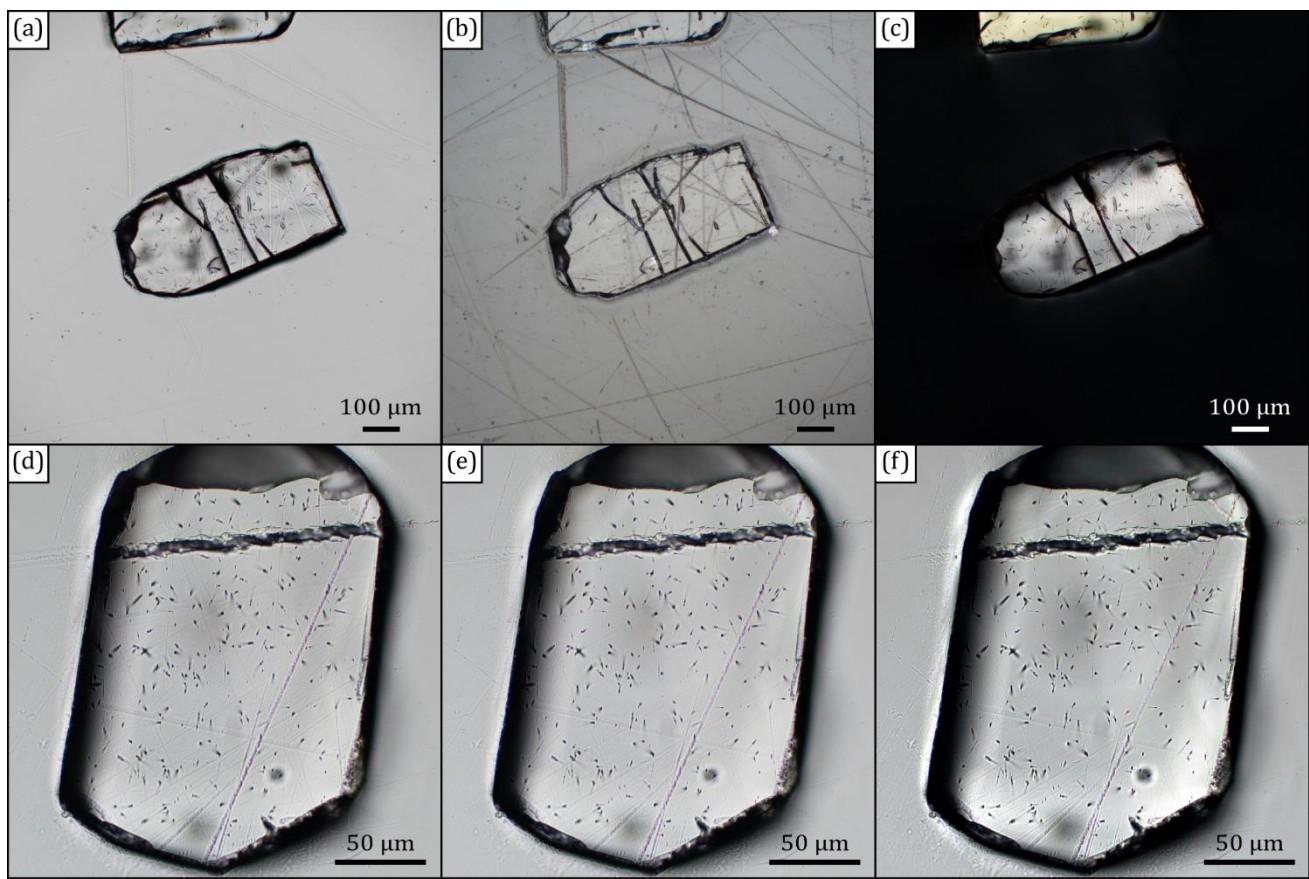

**Figure 7: Images created with the Nikon Eclipse Ni-E microscope and the DS-Ri2 camera. Subsets a–c illustrate different illumination conditions, with the 20x objective. a) Diascopic (DIA) illumination, b) episcopic illumination, c) circular polarisation. Subsets d–f illustrate the effect of camera exposure time (ET), analog gain (AG) and integration (I), with the 50x objective at DIA illumination. d) ET: 22 ms, AG: 1x, I: OFF, e) ET: 2 ms, AG: 9x, I: OFF, f) ET: 2 ms, AG: 1x, I: 8x. It can be observed that although an increase of the noise with higher AG the image quality remains within acceptable values, in combination with shorter exposure times and thus faster refresh rates. Integration causes a smoothening and decreases exposure time.**



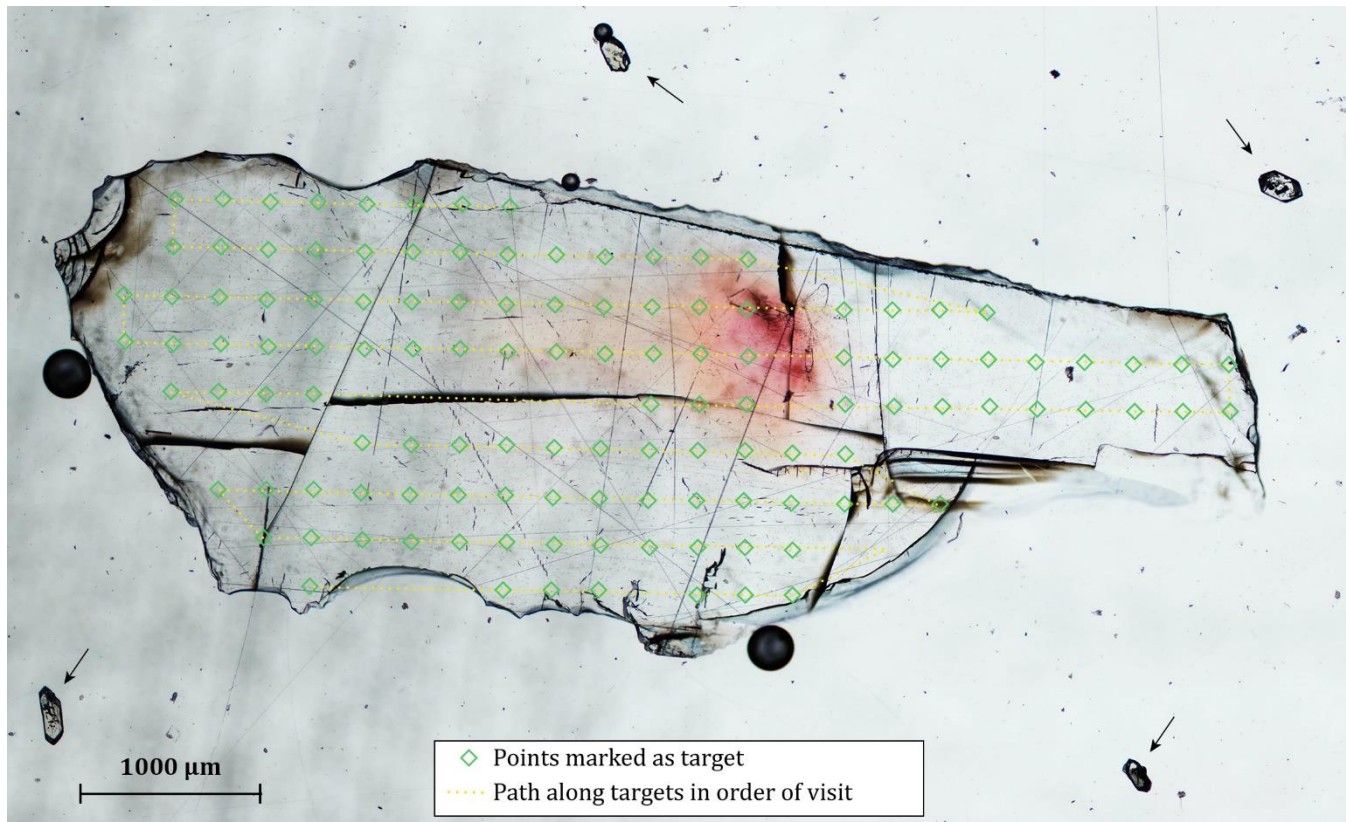

**Figure 8: Illustration of the XYZ Overview with a large image composed of stitched tiles. The image contains a thin slab of a Durango crystal and four co-embedded zircons as calibration points (indicated with arrows). The green diamonds are positions automatically generated by the TF_DUR protocol with automatic crystal detection and automatic inspection enabled. Points near the crack in the centre were not retained by the automatic inspection due to insufficient luminosity caused by the crack.**





**Figure 9: Condensed version of the TRACK*Flow*Chart demonstrating the major steps of the different protocols.**



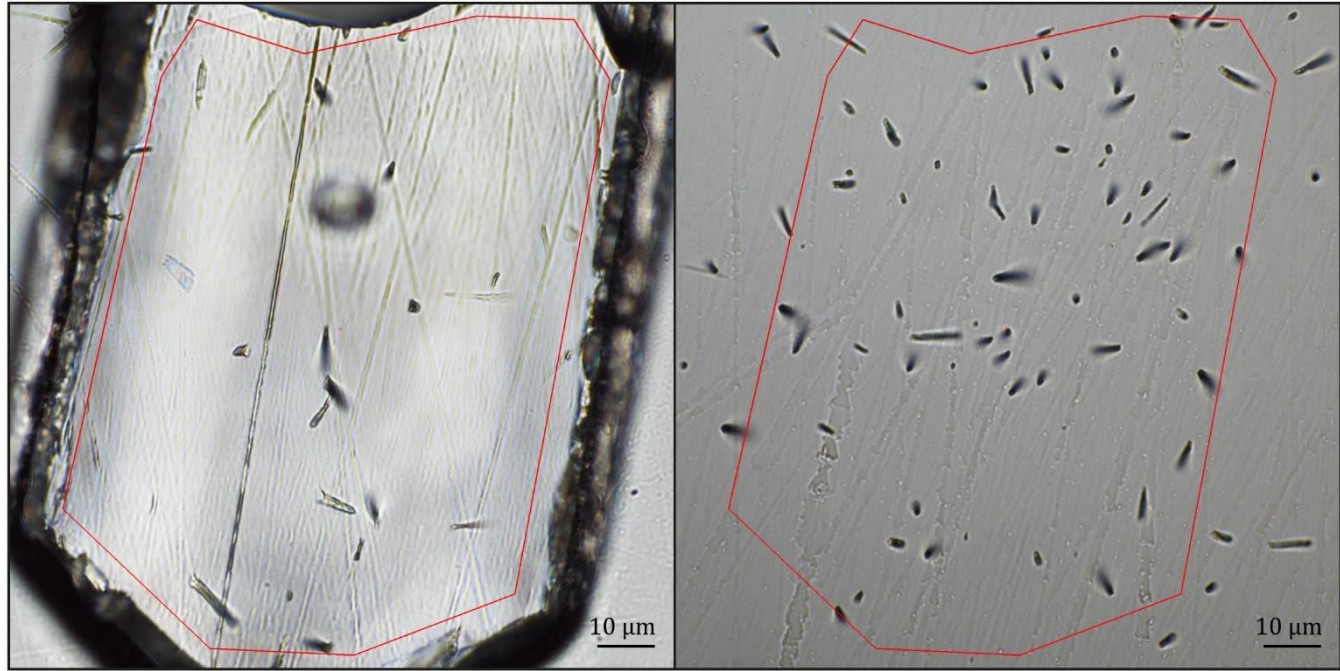

**Figure 10: Example of paired grain and its according induced tracks. Note that that the image is flipped horizontally and that a rotation of the image is also possible. The Region Of Interest (ROI) of the grain was copied to the ED.**





**Figure 11: Example of automatic D$_{par}$ analysis at 1000x magnification and 2x secondary optical magnification. Possible errors are a) false identifications, b) missed identification and c) inseparable double identification, which equals a false identification. In the histogram is shown that the false identifications (noise) do not pose a problem due to the large numbers of measurements. D$_{par}$ was calculated as the median.**