# Peer review of "Technical note: TRACKFlow, a new versatile microscope system for fission track analysis"

_Geochronology, 2019_

## Editor Comment (EC1) · Pieter Vermeesch (Editor) · 25 Oct 2019

The authors declare the following potential conflicts of interest:

- TRACKFlow is licensed from Ghent University to Nikon as a commercial product.

- Philippe Baert and Ana Clara Fernandes are employed at Nikon.

Note that all funding for this work was obtained from the Ghent University Special Research Fund.

---

## Short Comment (SC1) · 31 Oct 2019

In light of Twitter discussion of this paper as well as the fact that it is the first in the 'technical note' category and the first describing commercial rather than academic developments in geochronology, this seems like a good opportunity to clarify some of the Geochronology goals and policies.

First, we want to be clear that technical notes describing commercial products are not only permitted, but in fact encouraged. The purpose of the journal is to facilitate innovation in geochronology in all possible areas. Of course, when commercial products are involved it is important to keep in mind the distinction between factual descriptions

and marketing claims, and we will rely on both invited and unsolicited reviewers to help keep this clear through the open review process.

Second, with regard to conflict-of-interest statements, it is the responsibility of the authors to clearly and correctly disclose all conflicts. The journal can not, and does not, independently investigate the conflicting interests statements in all submitted papers. Again, we rely on the community of reviewers and readers to help with this through the open review process.

– Greg Balco on behalf of the Editors

---

## Referee Comment (RC1) · Hideki Iwano (Referee) · 4 Nov 2019

Referee comment on "Technical note: TRACKFlow, a new versatile microscope system for fission track analysis" by Gerben Van Ranst et al..

GENERAL

 The authors have developed a new microscope system for FT analysis, named TRACKFlow. The most notable improvements are as follows: 1) Obtaining an etched grain image can increase the degree of freedom of microscopic work and reduce stress. 2) Multiple samples can be placed on the stage and scanned easily without replacement. 3) This system has protocols that support both EDM and LAFT, and 4) has tools designed for large single crystal and U-doped glass external detectors. 5) In the case of EDM, the mirror image of ED can be converted into the same as the crystal. 6) Apatite D-Par can be automatically measured

 The system that is based on new technology realizes something that was not possible 20 years ago. Such improvement of the system can be expected to reduce human work and reduce mistakes. However, improvements (1) to (5) are automation before measurement. On the other hand, there is only improvement (6) related to measurement of D-Par, but it cannot be used for automated fission track counting.

 I have never used a standard TrackWORK system, but it is well known that the TrackWORK system has been used in many laboratories and made important researches. And automatic FT counting is also possible. I'm a Nikon user, using the Eclipse E1000 microscope installed twenty years ago. I admit that the performance of Nikon microscope is wonderful. In our laboratory, by combining with a touch sensitive monitor, we can superimpose a small red mark on each counted tracks and measure the number of tracks semi-automatically during FT counting on a PC monitor. This tool can reduce the burden on FT analysts (Danhara et al., 2007: Jour. Geol. Soc. Japan vol.113. 77-81.) Taking the above existing FT systems into account, it is expected that the authors focus on what's NEW improvement in your system. This MS may give readers an impression of a Nikon product catalog or something like that. That is not preferred. However, publishing a new and improved FT system is important information for FT researchers in the world and is suitable as a technical note.

 In general, the technical note (Short communication) should be shorter and more compact. This MS uses more than 6,500 words, so should be simplified further. I propose a major revision that at least halves the volume. I'd like to propose a simplified MS below only by deleting parts that were too detailed without technical corrections (because I am not a native. Sorry for it):

[revised manuscript text omitted]

(2,453 words)

SPECIFIC

1. Caption of Figure 1 should be corrected as follows:

Whole view of TRACKFlow system which consists of a motorized upright microscope with a color camera mounted and PC displays for track counting and storing data.

2. Figure 3 should be deleted to shorten the MS.

3. Figure 4 should be combined into one in Figure 1. Explanation for g) is missing.

4. Figure 5 should be deleted because it is not a new development.

5. Figures 6 and 7 should be deleted to shorten the MS.

6. Explanation for abbreviations in Figure 9 should be added.

7. DUR in text and Fig.9 seems strange and not general to readers. Large Single Crystal (LSC)?

With best regards,

Hideki Iwano

Iwano_hide@zeus.eonet.ne.jp

---

## Short Comment (SC2) · 21 Nov 2019

A review of:

"Technical note: TRACK*Flow*, a new versatile microscope system for fission track analysis"
Gerben Van Ranst, Philippe Baert, Ana Clara Fernandes, Johan De Grave

I do not believe that it is appropriate to publish this paper in this journal.

I see three major problems:

1.      Apart from a single example comparing some semi-quantitative image analysis derived Dpar measurements with some manual Dpar measurements in a single sample (a very simple procedure which does not require a sophisticated system), there is no information provided indicating how long it takes to do an analysis.   Since the authors claim an increase in FT counting productivity and increased laboratory throughput using this system, it is fair expect some evidence to back up the claims. For example, how many standard AFT samples can be counted and measured in a typical working day for example. Without such data, all the authors present are a series of unsubstantiated claims. There is no evidence that the features of the software described in the paper are actually advantages, and they may equally be impediments to efficient fission track determinations when implemented.

2.      It is simply not possible to provide a useful review of the software without actually using the system.

3.      Unfortunately, the article comes across as an advertisement for Nikon, and I would not be surprised if some of the paragraphs come from a brochure for the microscope system at the centre of the article.  I am surprised that since one of the authors is a Nikon employee that they don't acknowledge the possibility of a conflict of interest.  It raises a lot of questions, since the authors state that the software will be sold as a part of Nikon packages and will not be available in any other way.

Yours Sincerely

Dr. Ian R. Duddy.
Geotrack International Pty Ltd
37 Melville Road
Brunswick West   3055
Victoria
Australia
mail@geotrack.com.au

---

## Author Comment (AC1) · 25 Nov 2019

Reply to SC2: Ian Duddy, A review of:

"Technical note: TRACK*Flow*, a new versatile microscope system for fission track analysis"
Gerben Van Ranst, Philippe Baert, Ana Clara Fernandes, Johan De Grave

We thank Dr. Ian Duddy for his time to review our manuscript. We however disagree with his judgement, stating that "I do not believe that it is appropriate to publish this paper in this journal." and formulate a reply to the three posed major problems:

1. We acknowledge that we did not supply information on the time it takes to perform an analysis. We first emphasize that the main focus of this system is to provide a set of automated protocols for specific actions (e.g. scanning of a sample, glass ED or standard). Automated image *analysis* for fission track research is at this moment not included, as is mentioned as such, apart for $D_{par}$ measurements (which indeed does not require a sophisticated system, but which is more an additional module of TRACK*Flow*. As speed of counting of fission tracks is highly subjective, depending both on the (experience of the) analyst (or software package) and the sample, the relation to the system is marginal and thus not of importance for this note. We do also acknowledge that we did not provide the time for image acquisition. Foremost because of the fact that this acquisition time differs depending on the chosen protocol, the enabled options and on the linked computer. We suggest to solve this issue by providing information on the time it takes to acquire images of a regular sample and 100 spots on a Durango crystal with the prototype system. We will include such data in the revision of the manuscript.

2. We agree that this is a difficult problem which occurs in every occasion when new hardware-dependent software is discussed. Therefore we will provide demonstrations, for example at dedicated conferences or workshops. Furthermore, we suggest that the posting of a video may at least partially deal with this issue.

3. We regret that our manuscript comes across as an advertisement, as this was by no means the intention. This may be related to an overlap of the goal: informing about a product for scientific research. Indeed, in other academic–private ventures technical notes and brochures are sometimes published in parallel, whereas the former is formulated to *report* and *describe* the system and the latter is designed with a sales purpose (by a sales department). It is our understanding that the "technical note" section of this journal was intended to convey such a report and description, and we thought it was the ideal communication forum for it. Our aim in writing this manuscript was hence limited to inform and describe this system and how it can serve as a link in the long chain between sample preparation and thermal history reconstruction. In our opinion, this is the main goal of a technical note.
On the topic of conflict of interest we indeed first followed the example of broadly similar publications in the field, with authors affiliated to commercial companies. This issue was however resolved not long after the publication of the discussion paper (See EC1).

Yours sincerely,

**Gerben Van Ranst, M.Sc.**
D.Sc. student

**Department of Geology**
Laboratory for Mineralogy and Petrology
Campus Sterre, S8, Krijgslaan 281, B-9000, Ghent, Belgium
gerben.vanranst@ugent.be

---

## Referee Comment (RC2) · Andrew Gleadow (Referee) · 13 Dec 2019

Reviewer Comments on the "Technical Note: TrackFlow, a new versatile microscope system for fission track analysis" by Gerben Van Ranst, Phillipe Baert, Ana Clara Fernandes and Johan De Grave.

Potential conflict of interest declaration

This paper describes a new Nikon-based microscope system to assist with Fission Track (FT) analysis, which it is claimed, has a number of advantages, although being more limited in scope, than other currently available systems. The paper refers in

several places to the existing "TrackWorks suite from Autoscan systems" which is currently the only comprehensive system available for this purpose. To be strictly correct, TrackWorks is the microscope control and image capture package in the Fission Track Studio Suite, which includes the paired FastTracks image analysis and review package for offline data collection. This software suite is marketed by Autoscan Systems based on the Zeiss Axio-Imager microscope platform, but I need to declare at the outset that the software system, and the technical innovations encapsulated within it, have been entirely developed with our Research Group at the University of Melbourne. As the principal creator of this potentially competing system, I have raised this as a possible issue with the editor who subsequently confirmed that he would still like to have my review as I therefore provide as follows.

Comments on the paper:

A general point about this paper is that it has been submitted for publication as a Technical Note, which the Journal indicates should be 'several pages' in length. As noted by the other reviewers, however, the paper in its present form is much too long for this format, and indeed is longer than most of full research articles that are already published in Geochronology. By my estimate the paper as it is would come to about 13-16 published pages. It is up to the editors to decide how much latitude they will allow in the interpretation of 'several pages' but even the most generous interpretation suggests that this paper needs to be drastically reduced to no more than about 25-30% of its present length.

Another point that has also been raised by other reviewers is that the paper reads rather like an advertising brochure for Nikon, which should be modified to simply document the key features. I note that the original Conflict of Interest statement was deficient but has now been changed to reflect the fact that two Nikon employees are represented in the authorship. It is useful to learn that Nikon produces a fully motorised research microscope with specific features relevant to FT work, but these are then detailed as if they are unique to this system. In reality these features are common to essentially

all high-end motorised research microscopes. The same is true of the stages supplied with these microscopes which may be, and often are, equipped with Märzhäuser stages (page 4), as are most recent TrackWorks based systems. It is not clear why this was mentioned as an apparent point of differentiation. I think the specific features of the microscope used in this system need only the briefest summary, perhaps with reference to their website for additional details. This would greatly assist in reducing the length of the paper. Similarly, I think the background discussion of the basic FT method on pages 1 and 2 is largely unnecessary in a Technical Note and could be substantially reduced.

This brings me to the more substantive point that it is not obvious to me what is actually novel in this paper. At the simplest level the whole paper could be summarised as "We have duplicated some of the capabilities of TrackWorks using a Nikon microscope". At the top of page 3 of the contribution is described as "a novel microscope system developed and optimised for the fission track laboratory". The implication is that the discussion from that point on is describing a series of novel features, but in reality, almost all of these have been detailed in earlier publications and implemented in TrackWorks and FastTracks for years.

I am left with the impression that the only real point of novelty is the fact that the system is built within the Nikon operating environment. I have no problem with this being stated in an appropriately shortened Note, but I think that the authors need to be very clear about what they are claiming to be truly novel. Where they are simply duplicating specific innovations or features that have already been available in other systems, or described in previous publications, appropriate recognition and citation of that earlier work should be made. Much is made, for example, of the System Philosophy on page 3 which aims to obtain maximum efficiency by separating the microscopy from subsequent image-based analysis. This is definitely not new and has been the basis for the FT Studio suite since its inception in the early 2000s. This separation of tasks that frees up microscope time has been widely known since it was first presented publicly at

the European Thermochronology Conference in 2006 and detailed in subsequent publications. There are many other instances of re-inventing what has been done before, without proper attribution, such as the use of circular polarisation to discriminate apatite from epoxy, transforming coordinate systems, setting up a grid of counting points on a large area crystal, preserving images of tracks destroyed by later laser ablation, sending an email to the operator on completion etc. All of these, and more, are features of already existing systems. Even the name 'TrackFlow' is uncomfortably similar to 'TrackWorks'.

Similarly, great versatility is claimed for the new TrackFlow system because it is embedded within a broader generic microscope system. However, no evidence is given to support this assertion of greater versatility, and the Zeiss Axio-Imager system, which is the primary platform controlled by TrackWorks is also a generic high-level research microscope with multiple capabilities and broadly-based software options. Our Zeiss microscopes (under TrackWorks control), for example, are already regularly being used for other non-fission track applications in the earth sciences, so it is hard to justify the claim that this new system will be more versatile. These other applications include thin section analysis, imaging various other geological materials, 3D imaging of particles, pollen analysis, mineral grain characterisation, alpha track studies, and analysis of laser ablation pit dimensions, amongst others.

The very high-resolution camera (16 MPixel) utilised with the TrackFlow system is probably excessive for what is needed for FT work and likely to have significant drawbacks both in speed of operation and needlessly large file sizes. In FT analysis all images are usually captured at the highest magnifications (using a 100x objective), where the diffraction limited resolution of the optics determines that there is not much more than about 1 MPixel of useful information present. Some oversampling is useful to allow digital enlargement of the images, but 16MPixel is much more than is necessary and creates excessive demands on image storage and computational power for image analysis. The frame rate for such high-resolution cameras may also be quite poor which

Interactive
comment

can slow down image capture. Smaller format CMOS cameras can have full video frame rates at full resolution which greater facilitates imaging and storage, especially for high-throughput laboratories which this system seems to be aimed at, at least in part.

A couple of minor points are that the term 'focal plain' is used more than once, whereas the correct term should be 'focal plane'. Also, the term 'according' is used to describe the transformation between equivalent grain and induced track pairs in the External Detector Method. The terms 'corresponding' would make more sense.

Conclusion:

In summary, I think this paper could be acceptable for publication as a Technical Note in Geochronology, but only after major revision to drastically shorten it to the brief format anticipated for this kind of contribution. It is currently of the length of a Research article, but the content is not suitable for that format. Also, the tone of the paper needs to get away from the sense of it being an advertisement and focus much more on what is actually novel. Where 'new' features are described that already exist in closely similar form in other systems or have been documented in the literature, appropriate recognition and citation should be made.

Prof Andrew Gleadow School of Earth Sciences University of Melbourne Victoria, Australia E: gleadow@unimelb.edu.au
* * *

---

## Author Comment (AC2) · 13 Jan 2020

Dear Dr. Hideki Iwano,

Thank you for your review and your kind suggestions for the improvement of our manuscript. We understand that your main concern is the length of the manuscript for the category of technical note and the style, which gives it an impression of a Nikon product catalogue.

We will therefore try to reduce the length of the manuscript, keeping in mind your kindly provided suggestions, if this is also the conclusion of the Editor.

[Figure]

We have the following replies to your specific questions:

1. We will follow the suggested correction.

2. We will move Fig. 3 to Supplementary material, as an informative figure.

3. Combination of Figs. 1 and 4 would result in a reduction of the size of Fig. 4, which would reduce its readability and as such the information of the figure. We would therefore suggest to keep the two figures separate if the Editor agrees.

4. Figure 5f is a new development, whereas 5a–e are example images. We can adapt Fig. 5 to highlight the new aspect of box f.

5. We will move Fig. 6 to the Supplementary material. Fig. 7d–f gives important qualitative information on the recommended camera. We can however remove boxes a–c.

6. Explanations for abbreviations should and will be added.

7. We will follow the suggestion to replace DUR to the more general LSC (Large Single Crystal).

Best regards,

Gerben Van Ranst, on behalf of all co-authors

---

## Author Comment (AC3) · 23 Jan 2020

Dear Prof. Andy Gleadow,

Thank you for the review of our manuscript and for your constructive comments. We summarise your review to the following main issues:

1. The length of the manuscript is too lang and should be reduced.

2. Too many information on the microscope itself makes the manuscript resemble an advertising brochure.

[Figure]

3. It is not clear from the current manuscript what is novel and what already exists.

4. The Nikon DS-Ri2 camera is "overkill" for large magnifications and may be slow.

We formulate the following answers to these main concerns:

1. We will reduce the length of the note, taking into account the suggestions by reviewer Dr. Hideki Iwano and by you.

2. We will drastically reduce general information on the microscope and move a summary to the Supplementary data. We refer to the Nikon website / brochures where this is needed. This should already largely take away the style from the one of a brochure, which was unintended.

3. We first of all want to acknowledge the pioneering work of your Research Group at the University of Melbourne, which has revolutionised fission track research. We however want to negate the perception that TRACK*Flow* is a mere duplicate of TrackWorks, as is implied in your review. There are obviously a lot of similarities due to the simple fact that both packages have a same purpose. As they are both intended for the fission *track* laboratory, it is indeed no coincidence that the word "track" appears in both names. We do however point out that "*Flow*" carries the main weight, as the package is intended to ease the (work)flow for, amongst others, fission track research (the first module, thus "TRACK"). Herein, also the design of the mount and wellplate come into play for example. We also want to point out that we started the development of TRACK*Flow* because we wanted to invest in a system that is capable of being used for other tasks in our lab, and wanted to provide an alternative system for fission track imaging, yet from a different approach than Autoscan/TrackWorks. We further learned from other laboratories that similar needs as ours existed. For example, we included the possibility to select up to 10 homologous points (both primary and

secondary), of which primary points can be selected in a different (random) order on the mount and the ED. Secondary calibration points can be automatically picked randomly from the "target" apatites. The system prompts a warning if the transformation appears to be off. We implemented the possibility to fit multiple custom "irregular" (round, square) samples (1″ or smaller) on the stage while maintaining automatic stage movement (without operator input) to the centre of these mounts. Field diaphragm aperture is automatically adapted for thick (∼5 mm) mounts. These multiple mounts can be scanned in one run without operator intervention. An auto-exposure and *adaptive* autofocus (go to estimated surface, perform primary AF, retry with a larger interval when failed) is performed before each grain/spot is imaged. Several options, such as this AF or imaging (mount/ED) can be disabled. We focussed on task-specific protocols (flows), rather than step-by-step protocols. This contains e.g. automatic edge detection, grid generation and point inspection (epoxy, crystal edge, large crack) for large crystals (for example Durango), including imaging, without any other operator intervention except for starting the protocol and indicating the desired grid spacing. These differences are based on our experience with an older version (pre-2015) of TrackWorks and on discussions with other labs. It is thus well possible that what we believe is different, to the best of our knowledge, may be also available in newer versions of TrackWorks. We are however unable to make a detailed comparison as we do not own a license or the equipment for TrackWorks. We are therefore open for suggestions from you or the editor to improve the accuracy of our manuscript and are glad to add the necessary references. This clarification however does not resolve the fact that the impression may arise that we claim to introduce novelties which are not novel. This is by no means our intention. We believe that this impression may arise from the "generalised" writing style we adopted. For example, we state that 'higher efficiency can be obtained by separating image acquisition from analysis', which is indeed far from novel and is a principle that has been adopted before in other systems (e.g. TrackWorks) and

disciplines (e.g. Life Sciences). We propose that this issue can be resolved by discriminating more clearly between general principles, actual novelties inherent to TRACK*Flow* and information on the working of TRACK*Flow*. In the revised version we will take care to tune down claims of presenting a novel approach where this is not applicable. We emphasise that the main goal of our manuscript is to provide information about the TRACK*Flow* system, which itself is the novelty. We will however highlight what is novel, according to the best of our knowledge, as to make more clear which general principles have been applied. We further state that a direct comparison with similar systems, such as TrackWorks or other, non-commercialised systems, is beyond the scope of our manuscript.

4. We acknowledge that the DS-Ri2 camera has a more than sufficient resolution for the imaging of fission tracks at high ($100\times$ objective) magnifications, which can be seen as a disadvantage (e.g. raw image size) from a certain point of view. There are however also major advantages to this camera, which have led us to prefer this one over several other carefully tested camera's in 2016. First of all, we did not at all experience a speed drop, as is mentioned in your review. On the contrary, we selected this camera because of its smooth refresh and imaging, even at high magnifications. As we mention in the manuscript and as we demonstrate in Fig. 7, the speed can be modified e.g. by increasing analog gain, without compromising image quality whatsoever. Furthermore, this camera has an excellent signal/noise ratio due to its large pixel size. The camera also has excellent high dynamic range and colour reproduction, which becomes a strong advantage when imaging thin sections at different polarisation angles. The high resolution also proves its purpose when scanning at small magnifications. We emphasise that we selected this camera to meet the needs of a versatile system with many applications, which thus requires a top end camera. Finally, we state that the DS-Ri2 is the camera which we recommend and which we use on the prototype microscope. The user is free to select any other microscope camera

that is compatible with the Nikon NIS-Elements software suite.

Best regards,
Gerben Van Ranst, on behalf of all co-authors

---

## Author Response (AR1)

**Reply to the Associate Editor Decision**

Dear Associate Editor,

Thank you for your response and for your recommendations as to meet the requirements of the reviewers as to improve the quality and message of our manuscript. We tackled these concerns in the following manners:

1. **The manuscript is too long for a technical note.**

We reduced the length from 6262 words to 2249 words. This was mainly done by removing generalities from the introduction, replacing microscope aspects by a reference to the brochures and by removing supplementary modules in favour of the main modules. Some sentences were simplified.

2. **The text reads too much like an advertisement brochure.**

To deal with this issue we have removed most technical aspects inherent to the Nikon microscope and software and replaced this with a reference to the actual commercial brochure. We further removed most parts which are describing the Nikon system rather than the Nikon–TRACK*Flow* system, such as the nd2 window. Also as a consequence, the manuscript now has more focus on the new modules.

3. **Dr. Duddy made the point that "it is simply not possible to provide a useful review of the software without actually using the system."**

We understand this concern and want to deal with it in a way that is practically possible. Unfortunately, although we want to provide this in the future, the software is currently not in a state that allows us to create a trial version for a hands-on experience. We can however provide a demonstration of the system, for example through a video. For this review we have made a short video as to demonstrate one protocol. In the near future we can provide more of these video demonstrations.

4. **"There is no information provided indicating how long it takes to do an analysis."**

We included the durations for 20 points on a 'regular' sample and ~150 spots on a Durango crystal. We also mentioned the advantage that the system can work 24 hours, a suggestion for which we are grateful to the handling editor.

5. **Many of the claimed 'innovations' in TRACKflow have already been implemented in TrackWorks.**

We have addressed this issue by now clearly stating the common goals and functionality of these two systems. It is now clearly stated that the Melbourne group has developed the first complete and comprehensive system. Where we are aware of the same purpose/philosophy of the two systems, we state it as such. We further added a short paragraph on the general practice or the first implementations (e.g. of working with a motorised stage) as to make clear that these things were implemented in Nikon–TRACK*Flow* instead of invented and implemented. In dealing with this issue, we have conducted a search for the original references, which is sometimes complicated as it often concerns technicalities which were not the main topic of a publication and/or which are nowadays considered common practice, often lacking citation. As we do want to give credit to the inventors of these practices, we invite the reviewers to complement where they consider this necessary.

6. **The name 'TRACKFlow' is uncomfortably similar to 'TrackWorks'.**

We first of all changed the name to Nikon–TRACK*Flow*, as to make clear that this is a Nikon product. We also explain the rationale as focusing on the workflow for fission track research. We further argue that both the terms 'track' and '(A)FT' can be considered generic in the fission track community, considering their use in the names of multiple software packages.

**7. A 16Mpx camera is overkill for fission track counting using a 100x objective.**

We removed most technical details from the camera. However, since we carefully selected this camera as part of the system, we retain our recommendation on it. Nonetheless, we included the statement that any camera compatible with Nikon NIS-Elements can be used, with reference to the web page which displays the currently compatible cameras.

Yours sincerely,

Gerben Van Ranst, on behalf of all co-authors.

[revised manuscript text omitted]

---

## Referee Report (RR1)

Comments on the revised version of the paper by Van Ranst et al. – "Technical Note: Nikon-TRACK*Flow*, a new versatile microscope system for fission track analysis"

**General Comments:**

The authors have substantively dealt with all of the recommendations that I made in my first review, and in particular the paper is now of a length suitable for publication as a 'Technical Note'. I therefore now recommend that the manuscript be published subject to correction of one small error (two different names are used for the same 'LSC Tool' on page 4 - compare lines 3 'LSC' and 15 'Durango' tool). The same error is found in Fig 3 where the tool is called 'DUR' in the diagram and 'LSC' in the caption.  I also suggest that some very brief explanation is required on page 4, line 5, of just how the software 'automatically defines the bounding box'.  I have further made a number of minor suggestions to improve the English expression, below.

**Minor suggestions:**

*Page 1:*
Line 25: '…commercialised by Autoscan…'
Lines 33, 34 and 37:  'amongst others' is used three times in five lines, including twice in one sentence, which sounds very cumbersome and the meaning is not clear.  I suggest simply deleting the first occurrence in line 33 ('Both systems have the aim…'), changing the second one in line 34 to '…system, including to reduce schedule pressure.' and changing the last one in line 37 to '…they differ, amongst other things, in microscope brand…'.

*Page 2:*
Line 7:  'amongst others' appears once again and I suggest replacing it simply with 'includes' which is clearer and more direct.

*Page 3:*
Line 8: '…lenses in the case where a thin ED…'
Line 4:  should be 'copper', not 'cupper'
Line 29: '…retain the same order.'

*Page 4:*
Line 12: '…recoordination, a second scan of the same…'
Line 15: '…tool but optimised for the EDs…'
Line 16: 'The main difference is that … … automatically so that this action…'
Line 20: 'Coordinate transformation is therefore included in the glass tool for this purpose.'
Line 27: '24/7' is a colloquialism that is inappropriate for a scientific paper. It needs to be spelled out or said in some other way, perhaps '…24 hours a day, 7 days per week.' (probably unrealistic, as no microscope is going to be used that intensively), or simply '…around the clock.', or just '… day and night.', both of which are common ways of saying the same thing.
Line 33: '…so as to deal with a mismatch of the focal plane level with…'

Andrew Gleadow
7 April 2020

---

## Author Response (AR2)

**DEPARTMENT OF GEOLOGY**
LABORATORY FOR MINERALOGY AND
PETROLOGY

Gerben Van Ranst
*D.Sc. researcher*

E   gerben.vanranst@ugent.be
T   +32 9 264 45 68
M   +32(0)4 72 69 27 07

Campus Sterre, building S8
Krijgslaan 281
B-9000 Ghent
Belgium

The Editor
The Associate Editor
Geochronology

DATE
15 April 2020

PAGE

**Reply to the Editor Decision**

Dear Editor Greg Balco,
Dear Associate Editor Pieter Vermeesch,

Thank you for your response and for your final decision to accept our manuscript for publication in Geochronology. We are delighted that we can contribute to this new journal.

In the final version of our manuscript we have addressed all changes suggested by Prof. Gleadow. All references to the LSC tool are now uniform (including in the supplement). We also added a short explanation on how the system automatically detects the bounding box of large single crystals. We are also grateful to the spelling suggestions and corrections provided by Prof. Gleadow and as such applied them in our manuscript.

Yours faithfully,

Gerben Van Ranst, on behalf of all co-authors